

# Clinical characteristics and prognostic value of MEX3A mRNA in liver cancer

Dingquan Yang[1], Yan Jiao[2], Yanqing Li[3] and Xuedong Fang[1]

[1] Department of Gastrointestinal Colorectal and Anal Surgery, China-Japan Union Hospital of Jilin University, Changchun, Jilin, China
[2] Department of Hepatobiliary and Pancreatic Surgery, The First Hospital of Jilin University, Changchun, Jilin, China
[3] Department of Pathophysiology, College of Basic Medical Sciences, Jilin University, Changchun, Jilin, China

## ABSTRACT

**Background**. MEX3A is an RNA-binding proteins (RBPs) that promotes the proliferation, invasion, migration and viability of cancer cells. The aim of this study was to explore the clinicopathological characteristics and prognostic significance of MEX3A mRNA expression in liver cancer.

**Methods**. RNA-Seq and clinical data were collected from The Cancer Genome Atlas (TCGA). Boxplots were used to represent discrete variables of MEX3A. Chi-square tests were used to analyze the correlation between clinical features and MEX3A expression. Receiver operating characteristic (ROC) curves were used to confirm diagnostic ability. Independent prognostic ability and values were assessed using Kaplan–Meier curves and Cox analysis.

**Results**. We acquired MEX3A RNA-Seq from 50 normal liver tissues and 373 liver cancer patients along with clinical data. We found that MEX3A was up-regulated in liver cancer which increased according to histological grade ($p < 0.001$). MEX3A showed moderate diagnostic ability for liver cancer (AUC $= 0.837$). Kaplan–Meier curves and Cox analysis revealed that the high expression of MEX3A was significantly associated with poor survival (OS and RFS) ($p < 0.001$). Moreover, MEX3A was identified as an independent prognostic factor of liver cancer ($p < 0.001$).

**Conclusions**. MEX3A expression shows promise as an independent predictor of liver cancer prognosis.

Corresponding authors
Yan Jiao, jiaoyan16@mails.jlu.edu.cn, lagelangri1@126.com
Xuedong Fang, fangxd@jlu.edu.cn

# INTRODUCTION

Liver cancer is a malignant cancer with poor prognosis that is responsible for more than 780,000 deaths annually, making it the second most common cause of cancer-related mortality worldwide (*Bray et al., 2018*). Although liver cancer can be alleviated or cured through hepatectomy, orthotopic liver transplantation, and/or ablative procedures (*Hanouneh, Alkhouri & Singal, 2019*). More than 65% of patients fail to be cured and frequent recurrence contributes to poor survival. Predicting the overall 1-year survival rates for liver cancer have remained a challenge (*Galle et al., 2018*; *Hanouneh, Alkhouri & Singal, 2019*). New biomarkers that can predict liver cancer recurrence are urgently required to improve prognosis.

Liver cancer is influenced by post-transcriptional mechanisms that dynamically regulate protein expression (*Wong, Tsang & Ng, 2018*; *Goldstrohm, Hall & McKenney, 2018*). Cis-regulatory RNA elements and trans-acting factors (*Gerstberger, Hafner & Tuschl, 2014*; *Moore, 2005*) including RNA-binding proteins (RBPs) play an essential role in gene expression in cancer cells (*Masuda & Kuwano, 2019*). Recently, a group of RBPS termed MEX-3 RNA binding family member (MEX3) was identified in the nematode Caenorhabditis elegans (*Ciosk, DePalma & Priess, 2006*) and revealed one of the few RBPs with carcinogenic or tumor suppressor activity (*Kim, Hur & Jeong, 2009*; *Pereira et al., 2013a*).

MEX3 proteins are evolutionarily conserved RNA-binding proteins that consist of four homologous genes (MEX3A–D) (*Buchet-Poyau et al., 2007*; *Courchet et al., 2008*; *Pereira et al., 2013a*). They contained highly conservative one carboxy-terminal RING finger module and two K homology domains, the former mediating E3 ubiquitin ligase activity, the latter providing RNA-binding capacity (*Buchet-Poyau et al., 2007*). Available evidence implicates the MEX3 family in epithelial homeostasis, embryonic development, metabolism, immune responses and cancer, but the specific mechanisms of these effects require elucidation (*Pereira et al., 2013a*). MEX3A is a member of the MEX3 family (also known as RKHD4 or RNF162) that is expressed in endometrium tissue and the ovaries. MEX3A is a novel component of GW-182 or Dcp-containing bodies in mammals that represent cellular sites of mRNA degradation and the sequestration of non-translated transcripts (*Cougot, Babajko & Seraphin, 2004*; *Eystathioy et al., 2002*; *Sheth & Parker, 2003*).

MEX3A mRNA was recently shown to be overexpressed in Wilms tumors (*Krepischi et al., 2016*), gastric cancer (*Jiang et al., 2012*), bladder cancer (*Huang et al., 2017*), and bladder urothelial carcinoma (*Shi & Huang, 2017*). MEX3A promotes cell proliferation in bladder (*Huang et al., 2017*) and gastric (*Jiang et al., 2012*) cancer and shows potential as a biomarker to predict carcinogenesis (*Pereira et al., 2013a*). In this study, we analyzed the expression of MEX3A in liver cancer and assessed its clinicopathological potential. We further investigated the potential of MEX3A as an independent predictor of liver cancer prognosis.

## METHODS

### Clinical and RNA-Seq analysis

We downloaded the all RNA-Seq expression matrix from the Cancer Genome Atlas (TCGA) database and obtained MEX3A mRNA expression data from liver cancer vs. normal liver tissue using the matrix. We further obtained corresponding clinical and pathological information from TCGA database. The basic clinical data included age, gender, histological grade, stage, $T/N/M$ classification and vital status. MEX3A mRNA expression were estimated as log2(x+1) values and transformed RSEM normalized counts.

### Statistical analyses

We retrospectively analyzed all data using R (version 3.5.1) (*R Core Team, 2009*). We used non-parametric rank sum tests to analyze MEX3A mRNA expression levels according to different variables and boxplots were visualized. Wilcoxon rank sum tests were used to compare the two subgroups, including disease, age, gender and vital status. Kruskal–Wallis tests were used for the comparison of three or more subgroups, including clinical stage,

**Table 1** MEX3A mRNA expression and clinical characteristics in liver cancer patients.

| Characteristics | Number of pts(%) |
|---|---|
| Age | |
| <55 | 117(31.45) |
| >=55 | 255(68.55) |
| NA | 1(0.00) |
| Gender | |
| Female | 121(32.44) |
| Male | 252(67.56) |
| Histological_type | |
| Fibrolamellar Carcinoma | 3(0.8) |
| Hepatocellular Carcinoma | 363(97.32) |
| Hepatocholangiocarcinoma (Mixed) | 7(1.88) |
| Histologic_grade | |
| G1 | 55(14.75) |
| G2 | 178(47.72) |
| G3 | 123(32.98) |
| G4 | 12(3.22) |
| NA | 5(1.34) |
| Stage | |
| I | 172(46.11) |
| II | 87(23.32) |
| III | 85(22.79) |
| IV | 5(1.34) |
| NA | 24(6.43) |
| T_classification | |
| T1 | 182(48.79) |
| T2 | 95(25.47) |
| T3 | 80(21.45) |
| T4 | 13(3.49) |
| TX | 1(0.27) |
| NA | 2(0.54) |
| N_classification | |
| N0 | 253(67.83) |
| N1 | 4(1.07) |
| NX | 115(30.83) |
| NA | 1(0.27) |
| M_classification | |
| M0 | 267(71.58) |
| M1 | 4(1.07) |
| MX | 102(27.35) |
| Radiation_therapy | |
| No | 340(91.15) |
| Yes | 8(2.14) |
| NA | 25(6.7) |

**Table 1** (*continued*)

| Characteristics | Number of pts(%) |
|---|---|
| Residual_tumor | |
| R0 | 326(87.4) |
| R1 | 17(4.56) |
| R2 | 1(0.27) |
| RX | 22(5.9) |
| NA | 7(1.88) |
| Vital_status | |
| Deceased | 130(34.85) |
| Living | 243(65.15) |
| Relapse | |
| No | 179(55.94) |
| Yes | 141(44.06) |
| NA | 53(14.2) |
| MEX3A | |
| High | 117(31.37) |
| Low | 256(68.63) |

**Notes.**

NA, not available.

histologic grade and $T/N/M$ classification. We used the pROC package to draw ROC curves for the evaluation of MEX3A diagnosis through the calculation of AUC values and the measurement of optimal cutoff point to divide samples into high and low MEX3A expression groups (*Robin et al., 2011*). Further, correlation between clinical features and MEX3A expression group were analyzed through chi-square tests with Fisher's exact test.

To evaluate prognosis, Kaplan–Meier curves were used based on log-rank tests to compare differences in survival status, including overall survival (OS) and relapse-free survival (RFS) between the high and low MEX3A expression groups using the survival package in R (*Therneau, 1994*; *Therneau & Grambsch, 2000*). Univariate Cox analysis was used to select factors associated with prognosis, with calculations of hazard ratios (HRs) and 95% confidence intervals (95% CIs). Independent prognostic values of OS and RFS in the patients were determined through Multivariate Cox analysis. $P$-values $< 0.05$ were deemed statistically significant.

# RESULTS

## Clinical characteristics and RNA-Seq analysis

A total of 423 tissue samples with MEX3A mRNA expression data, including 373 liver cancer and 50 normal liver tissues were obtained from the TCGA. All patients were diagnosed with primary liver cancer. Corresponding patient demographic and clinical characteristics such as age, gender, histologic grade, TNM stage, vital status, and radiation therapy were obtained. All patient data is shown in Table 1.

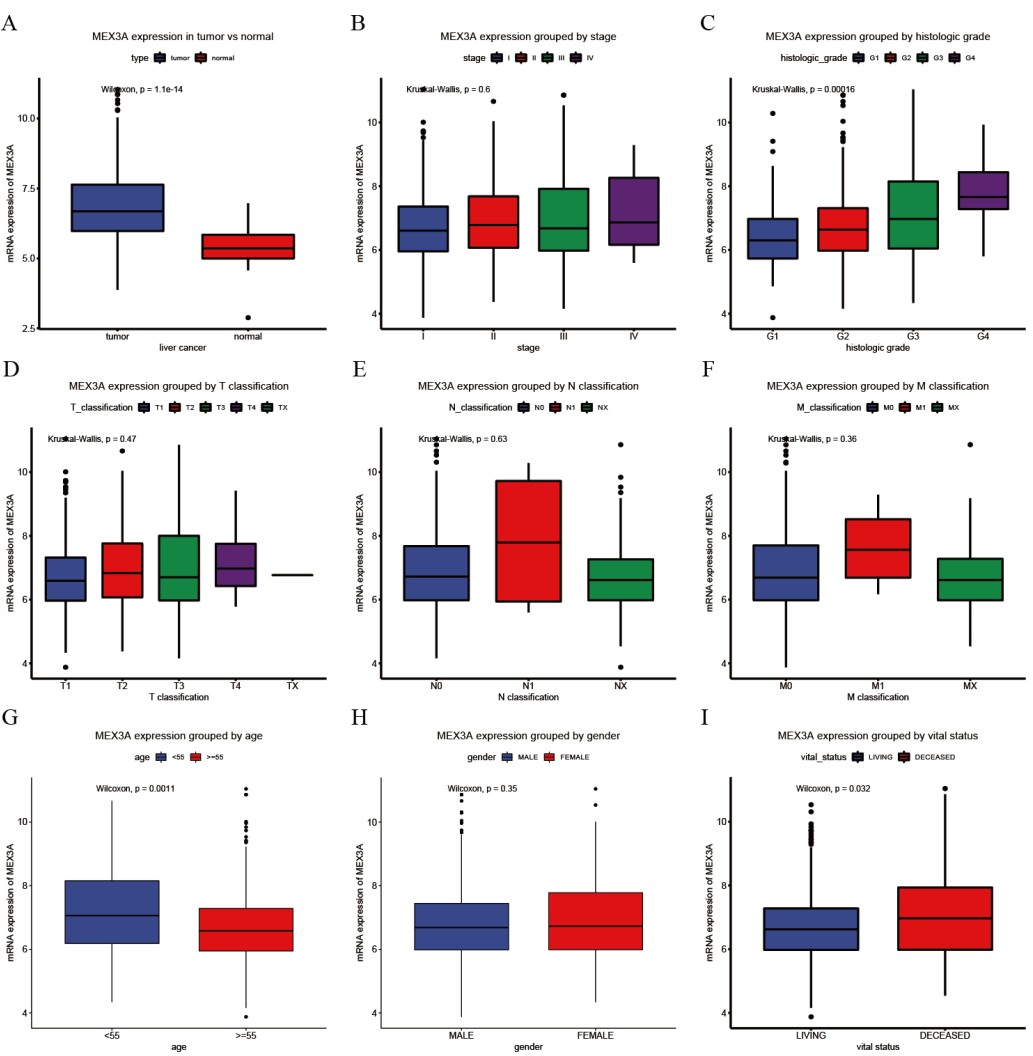

**Figure 1** **Assessment of the relationship between MEX3A mRNA expression and clinical characteristics (A–I).** Comparison of MEX3A mRNA expression in 373 cases of liver cancer and 50 normal liver tissues (A). Comparison of MEX3A mRNA expression according to clinical parameters: clinical stage (I, II, III and IV) (B), histologic grade (G1, G2, G3 and G4) (C), T classification (T1, T2, T3 and T4) (D), N classification (N0, N1 and NX) (E), M classification (M0, M1 and MX) (F), age (<55 and ≥55) (G), gender (male and female) (H) and vital status (I).

## MEX3A is highly expressed in liver cancer

In liver cancer, MEX3A mRNA expression level was significantly up-regulated compared to normal tissues ($p = 1.1e{-}14$; Fig. 1A) and increased with higher histological grades ($p = 0.00016$; Fig. 1C). MEX3A expression level was significantly associated with vital status ($p = 0.032$; Fig. 1I) and age ($p = 0.0011$; Fig. 1G).

## MEX3A as a liver cancer diagnostic.

ROC curve analysis showed that MEX3A had moderate diagnostic ability in patients with liver cancer (AUC=0.837; Fig. 2A). The diagnostic ability of MEX3A was comparable in all

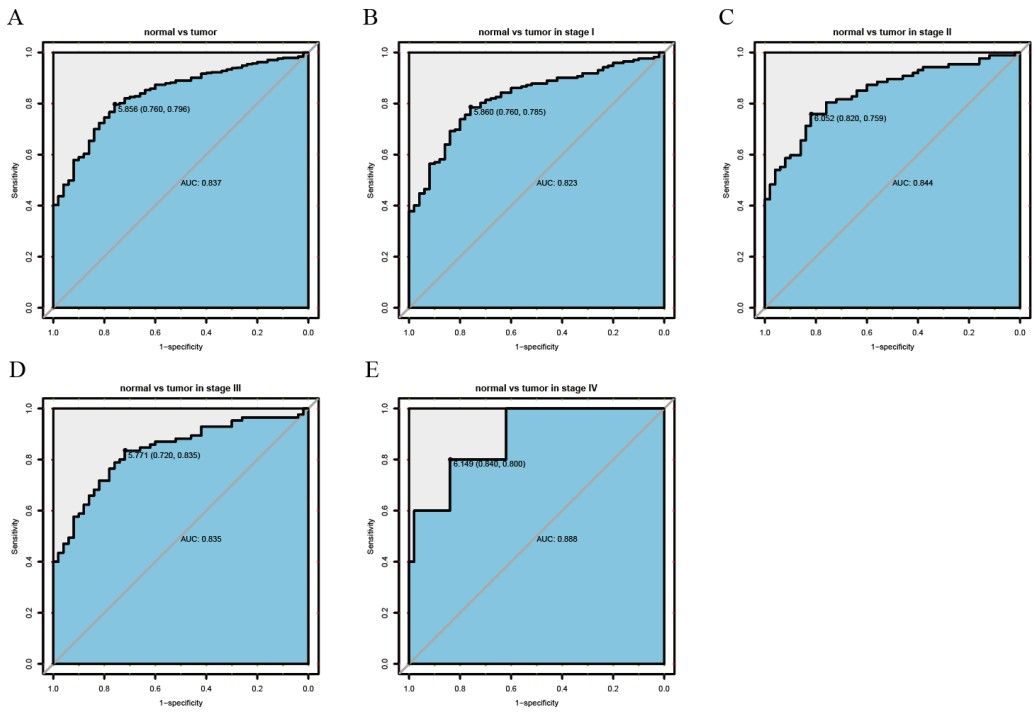

**Figure 2  ROC analysis of the sensitivity and specificity of MEX3A to assess liver cancer.** ROC curve for MEX3A expression in normal liver tissue and tumor tissue: AUC = 0.837 (A). I stage patients: AUC = 0.823 (B). II stage patients: AUC = 0.844 (C). III stage patients: AUC = 0.835 (D). IV stage patients: AUC = 0.888 (E). Abbreviations: AUC, area under the curve; ROC, receiver-operating characteristics.

clinical stages (stage I: AUC = 0.823, stage II: AUC = 0.844, stage III: AUC = 0.835, stage IV: AUC = 0.888; Figs. 2B–2E).

## Relationship between MEX3A expression and clinical characteristics

Figure S1 showed that the cutoff point was 7.266 when samples were divided according to high and low MEX3A expression. Chi-square tests were used to analyze clinical variables between the two groups, in which high MEX3A expression was associated with cancer related mortality ($p = 0.001$; Table 2). High MEX3A expression was also associated with a deterioration in liver tumor histopathology ($p < 0.001$; Table 2).

## MEX3A is an independent prognostic to evaluate the survival of liver cancer patients

Kaplan–Meier curves showed that patients with high MEX3A expression were more likely to have a poor OS ($p < 0.0001$; Fig. 3A). Further subgroup analysis showed that high MEX3A expression was associated with poor OS for all variables: stage I/II ($p = 0.0011$; Fig. 3B), stage III/IV ($p = 0.00022$; Fig. 3C), stage G1/G2 ($p < 0.0001$; Fig. 3D), stage G3/G4 ($p = 0.044$; Fig. 3E), male ($p < 0.0001$; Fig. 3F), female ($p = 0.0066$; Fig. 3G), younger ($p = 0.0026$; Fig. 3H), older ($p = 0.00022$; Fig. 3I). Through Univariate and Multivariate Cox analysis of OS, MEX3A (HR = 2.26, 95% CI [1.58–3.23], $p < 0.0001$)

**Table 2** Association between MEX3A expression and clinical characteristics in liver cancer patients.

| Clinical characteristics | Variable | No. of patients | MEX3A expression | | | | χ2 | *p*-value |
| --- | --- | --- | High | % | Low | % | | |
| Age | <55 | 117 | 51 | 43.59 | 66 | 25.88 | 10.8573 | **0.001** |
| | >=55 | 255 | 66 | 56.41 | 189 | 74.12 | | |
| Gender | Female | 121 | 43 | 36.75 | 78 | 30.47 | 1.1741 | 0.279 |
| | Male | 252 | 74 | 63.25 | 178 | 69.53 | | |
| Histological type | Fibrolamellar Carcinoma | 3 | 0 | 0 | 3 | 1.17 | 1.8006 | 0.547 |
| | Hepatocellular Carcinoma | 363 | 114 | 97.44 | 249 | 97.27 | | |
| | Hepatocholangiocarcinoma (Mixed) | 7 | 3 | 2.56 | 4 | 1.56 | | |
| Histologic grade | G1 | 55 | 10 | 8.7 | 45 | 17.79 | 20.0434 | **0.000** |
| | G2 | 178 | 48 | 41.74 | 130 | 51.38 | | |
| | G3 | 123 | 48 | 41.74 | 75 | 29.64 | | |
| | G4 | 12 | 9 | 7.83 | 3 | 1.19 | | |
| stage | I | 172 | 48 | 43.64 | 124 | 51.88 | 2.1216 | 0.512 |
| | II | 87 | 30 | 27.27 | 57 | 23.85 | | |
| | III | 85 | 30 | 27.27 | 55 | 23.01 | | |
| | IV | 5 | 2 | 1.82 | 3 | 1.26 | | |
| T classification | T1 | 182 | 49 | 41.88 | 133 | 52.36 | 4.9613 | 0.253 |
| | T2 | 95 | 35 | 29.91 | 60 | 23.62 | | |
| | T3 | 80 | 27 | 23.08 | 53 | 20.87 | | |
| | T4 | 13 | 6 | 5.13 | 7 | 2.76 | | |
| | TX | 1 | 0 | 0 | 1 | 0.39 | | |
| N classification | N0 | 253 | 86 | 73.5 | 167 | 65.49 | 3.4688 | 0.135 |
| | N1 | 4 | 2 | 1.71 | 2 | 0.78 | | |
| | NX | 115 | 29 | 24.79 | 86 | 33.73 | | |
| M classification | M0 | 267 | 88 | 75.21 | 179 | 69.92 | 2.0953 | 0.304 |
| | M1 | 4 | 2 | 1.71 | 2 | 0.78 | | |
| | MX | 102 | 27 | 23.08 | 75 | 29.3 | | |
| radiation therapy | NO | 340 | 106 | 98.15 | 234 | 97.5 | 0 | 1.000 |
| | Yes | 8 | 2 | 1.85 | 6 | 2.5 | | |
| Residual tumor | R0 | 326 | 97 | 84.35 | 229 | 91.24 | 5.1533 | 0.134 |
| | R1 | 17 | 7 | 6.09 | 10 | 3.98 | | |
| | R2 | 1 | 0 | 0 | 1 | 0.4 | | |
| | RX | 22 | 11 | 9.57 | 11 | 4.38 | | |
| Vital status | Deceased | 130 | 55 | 47.01 | 75 | 29.3 | 10.3281 | **0.001** |
| | Living | 243 | 62 | 52.99 | 181 | 70.7 | | |

Notes.

% represents the distribution of different clinical features in the single MEX3A expression group *P*-value in bold represent significant clinical significance ($p < 0.05$).

was identified as an independent risk factor for the prognosis of liver cancer along with T stage ($p < 0.0001$) and residual tumors ($p = 0.026$; Table 3).

Based on the OS, we further explored the connection between RFS and MEX3A expression, and found that high MEX3A expression was associated with poor RFS ($p < 0.0001$; Fig. 4A). RFS was related to the expression of MEX3A for some variables,

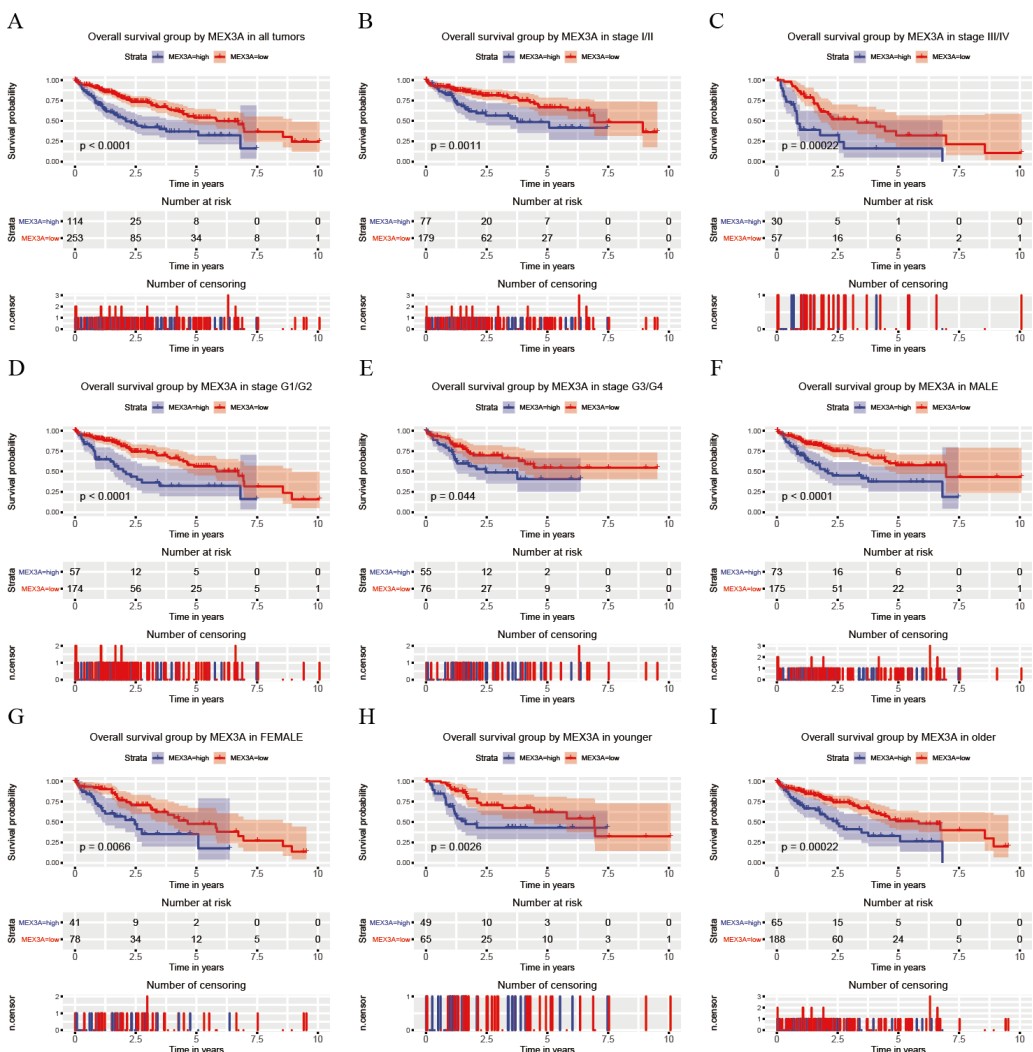

**Figure 3** **Analysis of OS between high and low expression groups of MEX3A according to the different clinical variables of liver cancer patients.** Kaplan-Meier curves of OS in all patients with liver cancer (A). Subgroup analysis was performed in stage I/II (B), stage III/IV (C), histological grade G1/G2 (D), histological grade G3/G4 (E), males (F), females (G), young patients (H) and old patients (I).

including stage I/II ($p = 0.013$; Fig. 4B), stage III/IV ($p = 0.00027$; Fig. 4C), stage G1/G2 ($p < 0.0001$; Fig. 4D), male ($p = 0.0023$; Fig. 4F), female ($p = 0.0049$; Fig. 4G), younger ($p = 0.0025$; Fig. 4H) and older ($p = 0.0047$; Fig. 4I). Univariate and Multivariate Cox analysis suggested that MEX3A (HR = 2.19, 95% CI [1.54–3.12], $p < 0.0001$) was an independent risk factor to evaluate the RFS for liver cancer along with T stage ($p < 0.0001$) and residual tumor status ($p = 0.024$; Table 4).

## DISCUSSION

Liver cancer is one of the deadliest tumors worldwide. The main risk factors for liver cancer include hepatitis B or C virus infections, the intake of aflatoxin, alcohol abuse

**Table 3  Relationship between clinical parameters, MEX3A mRNA expression and overall survival in liver cancer patients.**

| Parameters | Univariate analysis | | | Multivariate analysis | | |
|---|---|---|---|---|---|---|
| | Hazard ratio | 95%CI (lower~upper) | *P* value | Hazard ratio | 95%CI (lower–upper) | *P* value |
| Age | 1.00 | 0.69–1.45 | 0.997 | | | |
| Gender | 0.80 | 0.56–1.14 | 0.220 | | | |
| Histological type | 0.99 | 0.27–3.66 | 0.986 | | | |
| Histologic grade | 1.04 | 0.84–1.3 | 0.698 | | | |
| Stage | 1.38 | 1.15–1.66 | 0.001 | 0.86 | 0.69–1.07 | 0.163 |
| T classification | 1.66 | 1.39–1.99 | 0.000 | 1.88 | 1.48–2.38 | **0.000** |
| N classification | 0.73 | 0.51–1.05 | 0.086 | | | |
| M classification | 0.72 | 0.49–1.04 | 0.077 | | | |
| Radiation therapy | 0.51 | 0.26–1.03 | 0.060 | | | |
| Residual tumor | 1.42 | 1.13–1.8 | 0.003 | 1.33 | 1.03–1.71 | **0.026** |
| MEX3A | 2.29 | 1.61–3.26 | 0.000 | 2.26 | 1.58–3.23 | **0.000** |

Notes.

*P*-value in bold represent significant clinical significance ($p < 0.05$)

and non-alcoholic fatty liver disease (NAFLD) (*Zhang, Yang & Ericsson, 2019*). Due to the lack of effective diagnostic and prognostic evaluation methods, the mortality rates of liver cancer patients have gradually increased. New molecular markers that can guide prognosis and improve the survival rates of liver cancer patients are urgently required. Our team has devoted to exploring diagnostic and prognostic biomarkers in various cancers (*Hou et al., 2019*; *Jiao et al., 2018*; *Jiao et al., 2019a*; *Jiao et al., 2019b*; *Jiao et al., 2019c*; *Jiao et al., 2019d*; *Li et al., 2019*; *Sun et al., 2019*). In this study, MEX3A mRNA was identified as overexpressed in liver cancer tissue and could effectively evaluate the prognosis of liver cancer patients as an independent predictor. A strong correlation between high MEX3A expression and liver malignancy was also observed.

MEX3A is known to be upregulated in Wilms renal cancer (*Krepischi et al., 2016*), gastric cancer (*Jiang et al., 2012*), bladder cancer (*Huang et al., 2017*) and bladder urothelial cancer (*Shi & Huang, 2017*). These results are consistent with our finding that MEX3A mRNA is overexpressed in liver cancer ($p = 1.1e^{-14}$). Area under the ROC curves was 0.837, which provided evidence that MEX3A was a potential biomarker for liver cancer diagnosis. Interestingly, the expression of MEX3A was higher with increased histological grade ($p < 0.0001$), suggesting that MEX3A is related to tumor progression.

The molecular mechanism underlying the oncogenic effects of MEX3A remain poorly understood. *Huang et al. (2017)* found that MEX3A silencing significantly inhibits the proliferation of bladder cancer cells and promotes apoptosis. Jiang et al. similarly reported that MEX3A silencing delays the cell cycle progression of gastric cancer cells. MEX3A silencing significantly inhibited cell migration and anchorage-independent growth (*Jiang et al., 2012*). In colorectal cells, MEX3A is a stemness-related gene (*Barriga et al., 2017*; *Chatterji & Rustgi, 2018*; *Fernandez-Barral et al., 2019*) that acts as a repressive factor through controlling the expression of CDX2. CDX2 inhibits colorectal tumor cells growth, invasion, progression and migration and plays an essential regulatory role in intestinal homeostasis (*Bonhomme et al., 2003*; *Brabletz et al., 2004*; *Gross et al., 2008*; *Pereira et al.,*

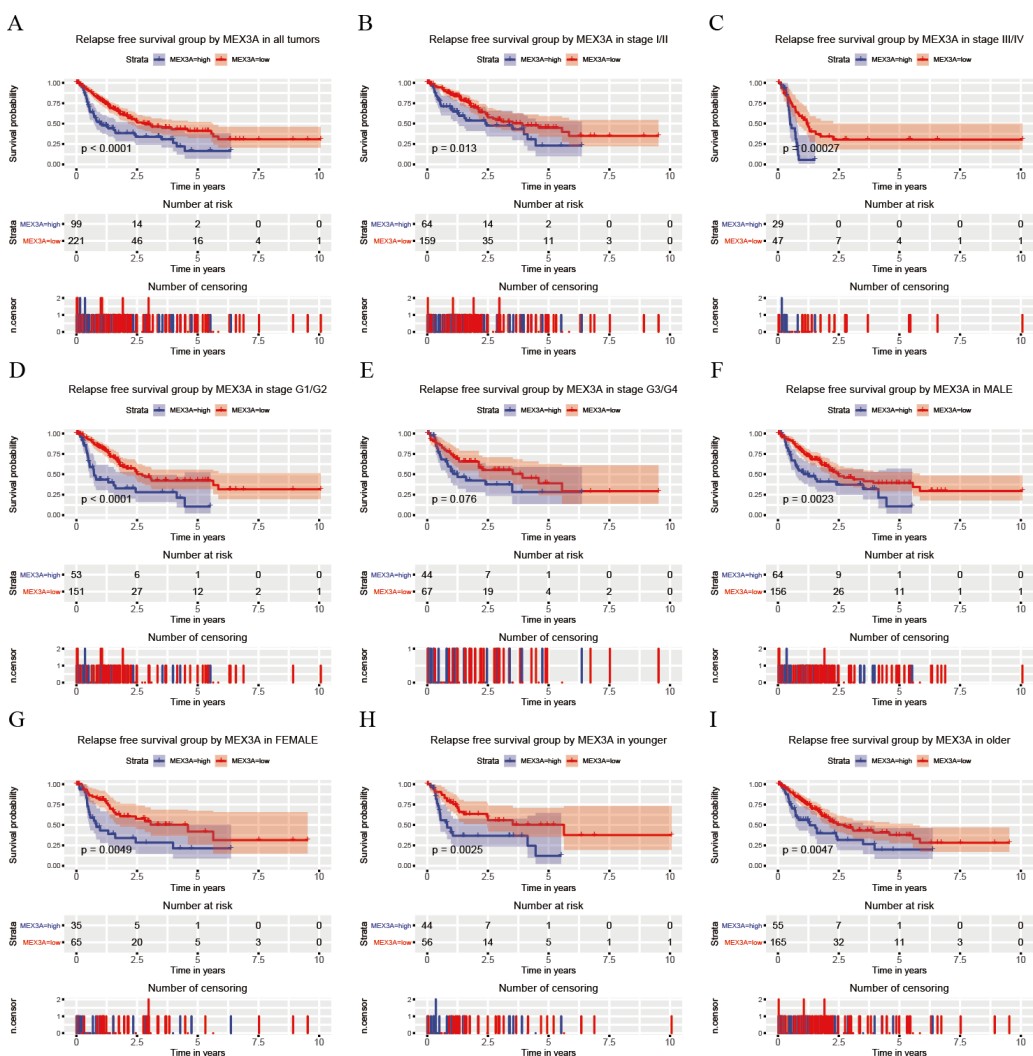

**Figure 4** **Analysis of RFS in high and low MEX3A expression groups according to the clinical variables of liver cancer patients.** Kaplan-Meier curves of RFS in all patients with liver cancer (A). Subgroup analysis was performed in stage I/II (B), stage III/IV (C), histological grade G1/G2 (D), histological grade G3/G4 (E), males (F), females (G), young patients (H) and old patients (I).

*2013b*). If MEX3A is overexpressed in colonic cell lines, cell polarity and differentiation become impaired leading to carcinogenesis (*Pereira et al., 2013b*). Combined with our findings, the role of MEX3A in cancer progression explains its clinical links to poor histological grade and poor patient prognosis in liver cancer. It is therefore necessary to explore the relationship between MEX3A and survival.

The MEX3 family shows promise as a biomarker for both cancer progression and prognosis (*Pereira et al., 2013a*). However, in studies by Huang et al. (*Shi & Huang, 2017*), no prognostic significance for cancer bladder urothelial carcinoma samples following MEX3A overexpression were observed. These results contrasted our findings and may highlight differential roles of MEX3A in cancer progression according to cancer-type.

**Table 4 Relationship between clinical parameters, MEX3A mRNA expression and relapse-free survival in liver cancer patients.**

| Parameters | Univariate analysis | | | Multivariate analysis | | |
|---|---|---|---|---|---|---|
| | Hazard ratio | 95%CI (lower~upper) | P value | Hazard Ratio | 95%CI (lower–upper) | P value |
| Age | 0.90 | 0.63–1.28 | 0.550 | | | |
| Gender | 0.99 | 0.7–1.41 | 0.966 | | | |
| Histological type | 2.02 | 0.66–6.24 | 0.220 | | | |
| Histologic grade | 0.98 | 0.8–1.21 | 0.883 | | | |
| Stage | 1.66 | 1.38–1.99 | 0.000 | 1.13 | 0.87–1.46 | 0.358 |
| T classification | 1.78 | 1.49–2.12 | 0.000 | 1.69 | 1.29–2.21 | **0.000** |
| N classification | 0.97 | 0.67–1.4 | 0.874 | | | |
| M classification | 1.17 | 0.79–1.74 | 0.432 | | | |
| Radiation therapy | 0.74 | 0.26–2.16 | 0.584 | | | |
| Residual tumor | 1.28 | 1.01–1.61 | 0.042 | 1.32 | 1.04–1.67 | **0.024** |
| MEX3A | 2.05 | 1.46–2.9 | 0.000 | 2.19 | 1.54–3.12 | **0.000** |

**Notes.**

$P$-value in bold represent significant clinical significance ($p < 0.05$).

Of note, upon assessment of each sub-variable group, Kaplan–Meier curves revealed that the liver cancer patients with high MEX3A expression had a poor OS. RFS were evaluated according to MEX3A expression and showed a similar relationship with OS, excluding G3/G4 group. This highlights the unique superiority of MEX3A expression for the assessment of liver cancer survival. In particular, as shown in Tables 3 and 4, MEX3A may represent a prognostic marker for liver cancer survival (OS and RFS) under the strong confounding effects of clinicopathological features, providing useful references for clinicians to aid the development of individualized patient's treatments.

In summary, this is the first study to report the association between MEX3A mRNA and the clinical characteristics and survival of liver cancer patients. MEX3A has great potential to predict the prognosis of liver cancer patients. Future studies should explore the mechanisms by which MEX3A promotes liver cancer *in vivo* and *in vitro*. Further clinicopathological information and corresponding clinical tissue samples should be obtained to further validate these findings and to establish MEX3A as a novel prognostic for patients with liver cancer.

## CONCLUSIONS

This is the first study to investigate the expression of MEX3A mRNA in liver cancer, revealing its association with specific clinical features. Moreover, our results indicate that MEX3A plays a significant role in the prognosis of liver cancer and can be used as an independent factor to predict liver cancer progression.

### Funding

The authors received no funding for this work.

## Competing Interests

The authors declare there are no competing interests.

## Author Contributions

- Dingquan Yang analyzed the data, performed the experiments, prepared figures and/or tables, authored or reviewed drafts of the paper, and approved the final draft.
- Yan Jiao and Xuedong Fang conceived and designed the experiments, performed the experiments, authored or reviewed drafts of the paper, and approved the final draft.
- Yanqing Li analyzed the data, prepared figures and/or tables, and approved the final draft.

## Data Availability

The raw data is available from the TCGA: https://xenabrowser.net/datapages/?cohort=TCGA%20Liver%20Cancer%20(LIHC)&removeHub=https%3A%2F%2Fxena.treehouse.gi.ucsc.edu%3A443%22.

## Supplemental Information

Supplemental information for this article can be found online at http://dx.doi.org/10.7717/peerj.8252#supplemental-information.

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
