# Peer review of "Clinical characteristics and prognostic value of MEX3A mRNA in liver cancer"

_PeerJ, doi:10.7717/peerj.8252_

## Round 0.1 · original submission · Major Revisions

All critiques of all the reviewers should be addressed and the manuscript should be revised accordingly.

Reviewer 1 ·

Basic reporting

The manuscript needs significant improvement on the language throughout.

Experimental design

Methods are not sufficiently described, for instance do not refer the normal samples that are used (number, etc).
RFS cannot be performed for stage IV patients, so these cases must be excluded from this analysis.

Validity of the findings

No comment

Reviewer 2 ·

Basic reporting

Yang et al. analyzed the RNA expression of MEX3A in TCGA data base and conclude that MEX3A expression is correlated to the disease development of live cancer and could be used as a prognosis markers. This is an interesting study and could be further improved.

1, The English need to be further edited by native speaker. For example, line 64, “The low overall 1-year survival for liver cancer confuse us all the time.” Line 83, “it’s an essential future challenge that is evaluating its potential as a biomarker for progression or prognosis of the disease and exploring its contribution to carcinogenics”

Experimental design

3, Different statistics methods are mixed in Figure 1. Why and how are they selected?

Validity of the findings

4, Fig.1, all fonts are too small. Please also clearly labelled all the single panels using A, B, C,… and provide clear description of each small panels in the figure legends. For comparison between multiple groups, it’s important to clearly label which two groups are compared.

5, Fig. 2, 3 and 4, please clearly labelled all the single panels using A, B, C,… and Figure legends as requested as above.

Reviewer 3 ·

Basic reporting

See the general comments

Experimental design

See the general comments

Validity of the findings

See the general comments

Additional comments

In this manuscript, Yang and co-workers reported the relationship between MEX3A mRNA expression level and clinical characteristics in liver cancer patients. The statistical results indicate that the MEX2A expression level can be used as an independent factor to predict the prognosis of liver cancer. This manuscript is somehow poorly written. There are numerous grammatical and stylistic errors as well. Some of the language mistakes cause the interpretation of experimental results confusing and ambiguous. I strongly recommend the authors to fix the errors and carefully proof-read before its resubmission to the journal. Overall, this is an important study and the findings are interesting. However, the lack of details of data collection and the ambiguous interpretations of the statistical results make it less convincible. Besides the language, the questions/suggestions for this research are listed below.

1. The definition of MEX3A expression level is not clear. The authors mentioned that the MEX3A RNA-Seq expression data were obtained from the TCGA without any further description that how the expression level was measured. For example, the y-axes of boxplots in Figure1 are labeled as mRNA expression of MEX3A, but the values are meaningless without explanation. Please specify what method was used to measure the MEX3A expression level and describe the expression level data in the method section with more details.

2. The MEX3A expression level is classified into two groups as high and low. The authors claimed that "measurement of reasonable median value that served as a cutoff to divide the samples into high and low MEX3A expression groups". However, the cutoff point (median value) is not reported in the manuscript.

3. The experimental method section is too brief, especially for the data collection part. More details are required to make the results solid. For example, based on the plots in the manuscript, it looks like the MEX3A mRNA expression level was measured in both normal tissue and tumor and statistically compared. However, no information is mentioned in the method section.

4. The interpretation of the statistical results is ambiguous and confusing. For example, "The MEX3A mRNA expression level was significant(ly) upregulated in liver cancer". Does it mean that the MEX3A expression level in the tumors is higher than the normal tissues, or the MEX3A expression level in patients with liver cancer is higher than the normal people? "The results of ROC curve reveal that MEX3A mRNA had moderate diagnostic ability in patients with live cancer". However, the ROC curve only indicates that the tissue with high MEX3A expression is more likely to be a tumor. So I don't see the potential of MEX3A as a biomarker to diagnose the liver cancer.

5. The percentage data reported in table 2 are confusing without description. Please explain in the table caption.

Reviewer 4 ·

Basic reporting

This manuscript focusses on role of RNA binding protein on cancer, specifically on liver cancer. Authors have chosen. The is manuscript was well written with professional language. However, the authors can improve sentence formation. Authors. Literature work is thorough with up to date references. Introduction part was very well written with background information of cancer and RNA binding proteins. Tables and figures are represented well with clear legends, marking, and labels. Authors have shown that the expression of MEX3A mRNA can be used as a prognostic molecule for liver cancer.

Experimental design

Manuscript meets the aims and scope of the journal and experimental designs are well presented. Authors have used adequate methodology to verify the differences in the expression of MEX3A in patients with multiple differences not only in age but also in cancer stage, gender, Overall survival etc. Authors have used a large data sets for their analysis. Authors are encouraged to use other profiling techniques such as Immunohistochemistry and metabolite profiling to identify potential targets for cancer studies.

Validity of the findings

Results part was written well and study shows original results from designed experiments. Some of the discussion part can be easily kept into introduction. Authors attempt to do comparative study with other similar work was appreciable but the discussion can be shortened and focus more on their findings. When discussing some of the results, authors should cite particular figure or table in the discussion. It will be interesting to learn about what is the mechanism of MEX3A and also studies based on knockouts of MEX3 family proteins. The OS and RFS results are encouraging showing the significance of MEX3A mRNA expression in relation to survival in liver cancer patients.

Additional comments

This manuscript was focused on investigating the role of MEX3A mRNA in liver cancer and how the expression variation can be used for prognosis. Introduction part was very well writtern. Experimental procedures and results section were written well. However, Care needs to be taken in terms of sentence formation and grammar. Some areas of the discussion part should be placed into introduction. Overall the manuscript was well written with minor revisions required. Please see comments from the attachment. Authors should also provide some future work on how they will use this molecule to future studies on liver cancer.

---

## Round 0.2 · accepted · Accept

As you can see, all reviewers were completely satisfied with your replies to their questions and with the manuscript revision.

Reviewer 1 ·

Basic reporting

No comment.

Experimental design

No comment.

Validity of the findings

No comment.

Additional comments

The clarity of the manuscript improved significantly with revision. The message is novel and relevant. The study is well performed with a relevant number of samples.

Reviewer 2 ·

Basic reporting

The manuscript clearly reported the potential of MEX3A in liver cancer diagnosis.

Experimental design

The work is well designed.

Validity of the findings

Data is prepared is a standard way.

Additional comments

The authors well addressed my concerns.

Reviewer 3 ·

Basic reporting

See the general comments

Experimental design

See the general comments

Validity of the findings

See the general comments

Additional comments

In this resubmission, the authors added more details in the method section. The detailed description of data collection and statistical methods make the results much clearer. The statistical results clearly indicate that MEX3A mRNA expression associated with the survival rate of liver cancer patients and it could be used as an independent predictor of liver cancer prognosis. The conclusion is more convincing than the original version. Overall, the authors addressed the questions from reviewers properly, and the resubmission is well written. I recommend that the paper should be accepted for publication in its present form.